# Exploration of the Regulatory Pathways and Key Genes Involved in the Response to Saline–Alkali Stress in *Betula platyphylla* via RNA-Seq Analysis

**DOI:** 10.3390/plants12132435

**Published:** 2023-06-24

**Authors:** Jukun Xue, Hu Sun, Xuemei Zhou, Huiyan Guo, Yucheng Wang

**Affiliations:** 1Department of Life Science and Technology, Mudanjiang Normal University, Mudanjiang 157011, China; swxxjk@126.com (J.X.); fxzxm0818@163.com (X.Z.); 2College of Forestry, Shenyang Agricultural University, Shenyang 110866, China; sh1837805814@163.com; 3The Key Laboratory of Forest Tree Genetics, Breeding and Cultivation of Liaoning Province, Shenyang Agricultural University, Shenyang 110866, China

**Keywords:** *Betula platyphylla*, saline–alkali stress, *WRKY70*, *NAC9*, ROS scavenging ability

## Abstract

The pH of saline–alkali soil is high because of carbonate salts, and the deleterious effects of saline–alkali soil on the growth of plants are greater than those of saline soil. Few studies have examined the saline–alkali tolerance of *Betula platyphylla* at the molecular level. To clarify the regulatory mechanism underlying saline–alkali tolerance in *B. platyphylla*, RNA sequencing analysis of *B. platyphylla* seedlings treated with NaHCO_3_ was conducted. Differences in gene expression in the roots of *B. platyphylla* seedlings under saline–alkali stress (induced via NaHCO_3_) for 3 h and 6 h were characterized, and a total of 595 and 607 alkali stress-responsive genes were identified, respectively. Most differentially expressed genes were involved in stress, signal transduction, secondary metabolic process, regulation of jasmonic acid, and the abiotic stimulus signaling pathway. The single nucleotide polymorphism loci in the differentially expressed genes were associated with the alkaline-salt tolerance in birch germplasm. In addition, birch plants overexpressing *WRKY70* and *NAC9* were obtained using the *A. tumefaciens*-mediated transient transformation method, and these two genes were found to play key roles in saline–alkali tolerance. Additional study revealed that *WRKY70* and *NAC9* can increase resistance to saline–alkali stress by enhancing reactive oxygen species scavenging and inhibiting cell death in birch plants. The results of this study enhance our understanding of the saline–alkali stress tolerance of *B. platyphylla* at the molecular level, and provide several key genes that could be used in the breeding of birch plants in the future.

## 1. Introduction

The total area of saline–alkali land globally has reached 954 million hectares, and the area of saline–alkali land is increasing by 10% annually [1]. Saline–alkali soil is mainly caused by sodium carbonate (Na_2_CO_3_ and NaHCO_3_), which increases the pH and Na^+^ concentration, decreases the water potential, and results in drought [2,3,4,5]. Thus, land saline alkalization has become an increasingly serious problem, as it can decrease the osmotic potential of soil, disrupt ion homeostasis, disturb physiological processes, inhibit the growth and development of plants, lead to declines in quality and yield, and even result in the death of plants [6,7]. Therefore, study of the mechanisms underlying the responses of plants to saline–alkali stress will aid the use of biotechnology to breed plants with improved saline–alkali tolerance. The root tissues of plants are the main organ responsible for the perception of several types of water-limiting stress, such as salinity and alkali stress [8,9]. Furthermore, the sensitivity of the roots to stress often limits the growth and development of the entire plant. Thus, the roots provide an excellent organ for clarifying the molecular mechanism underlying the saline–alkali stress tolerance of plants.

The mechanisms regulating stress resistance in plants involve the expression of multiple genes and the interactions among them, rather than individual genes [10,11]; thus, achieving a complete understanding of abiotic stress-tolerance mechanisms requires study of gene expression patterns. RNA sequencing (RNA-seq) analysis can rapidly provide information on gene expression and identify novel genes that mediate responses to various physiological processes in plants.

*Betula platyphylla* is a common pioneer tree in eastern Asia that plays a key role in maintaining ecosystem stability and forest regeneration. It is also widely used in architecture, furniture, and paper production [12,13,14]. Various aspects of the saline–alkali tolerance of plants have been analyzed in previous studies. However, plant saline–alkali tolerance is complex and controlled by a cluster of genes; it also varies among plant species and cultivars and involves various physiological, biochemical, and metabolic pathways. Study of the saline–alkali resistance mechanism of *B. platyphylla* could provide new insights into its saline–alkali tolerance and have important practical implications.

Here, transcriptomes of *B. platyphylla* roots that had been exposed to saline–alkali stress for different lengths of time were analyzed using high-throughput RNA-seq technology. The transcriptional pathways and some differentially expressed genes (DEGs) were identified, and the roles of key genes in saline–alkali stress tolerance were analyzed. Our findings provide valuable information that will aid future efforts to enhance the resistance of plants to saline–alkali stress via genetic engineering.

## 2. Results

### 2.1. Quality Control of the RNA-Seq Data

To obtain an overview of the transcriptome of *B. platyphylla* roots in response to saline–alkali stress, nine cDNA libraries were constructed using Illumina sequencing technology. These cDNA libraries were from *B. platyphylla* roots that had been exposed to 0.2 M NaHCO_3_ for 3 h (N3-1, N3-2, and N3-3) and 6 h (N6-1, N6-2, and N6-3); the control plants (ck-1, ck-2, and ck-3) were treated with fresh water for 6 h (three biological replicates for each treatment). A total of 66.45 Gb clean data were obtained, with each sample having at least 6.22 Gb. The GC content of each sample was approximately 46%, and the Q30 ranged from 92.8% to 93.95% (Appendix A). The clean reads of each sample were mapped against the *B. platyphylla* reference genome, with alignment rates ranging from 92.28% to 93.08% (Appendix A). Overall, these results suggest that the sequencing data were of high quality.

### 2.2. Identification of DEGs

The DEGs were identified by RNA-seq analysis. A total of 595 and 607 DEGs were identified following exposure to NaHCO_3_ stress for 3 h and 6 h, respectively (Table 1). A total of 290 and 305 up-regulated and down-regulated DEGs, respectively, were identified in the 3 h NaHCO_3_ treatment vs. the control comparison group; a total of 199 and 408 up-regulated and down-regulated DEGs, respectively, were identified in the 6 h NaHCO_3_ treatment vs. the control comparison group. We made a Venn diagram to analyze the overlap in DEGs across plants treated with NaHCO_3_ for different lengths of time; the results revealed that 188 co-expressed genes were differentially expressed in the 3 h and 6 h treatment groups (Figure 1).

### 2.3. GO Enrichment Analysis and Functional Annotation of DEGs

To clarify differences in gene expression related to biological pathways, Gene Ontology (GO) annotations of the DEGs in the three categories of biological process, cellular component, and molecular function, were obtained (Figure 2). In the biological process category, cellular, metabolic, and single-organism processes, response to stimulus, and signaling were highly enriched. In the cellular component category, membrane, membrane part, cell, cell part, and organelle were highly enriched. In the molecular function category, binding, catalytic activity, nucleic acid binding transcription factor (TF) activity, and transporter activity were highly enriched.

The clusters of orthologous groups (COG) classification statistics for DEGs revealed that signal transduction mechanisms, general function prediction only, carbohydrate transport and metabolism, and secondary metabolite biosynthesis, transport, and catabolism were highly enriched (Figure 3).

### 2.4. Hierarchical Clustering Analysis of DEGs

DEGs encoding CYP82C2, cinnamyl-alcohol dehydrogenase (CAD) genes, and DREB, WRKY, NAC, ERF, and bHLH TFs, were involved in response to stress, signal transduction, secondary metabolic process, regulation of jasmonic acid, and the abiotic stimulus signaling pathway (Appendix A). We performed a hierarchical clustering analysis of the expression patterns of genes related to the saline–alkali signal pathway. The expression patterns of most genes were similar in the 3 h and 6 h NaHCO_3_ treatments and the control (Figure 4). According to the hierarchical clustering analysis, the genes could be roughly classified into five groups. The first group included genes with up-regulated expression in the 3 h NaHCO_3_ treatment but down-regulated expression in the 6 h NaHCO_3_ treatment. Genes with slightly up-regulated expression in the 3 h NaHCO_3_ treatment and highly up-regulated expression in the 6 h NaHCO_3_ treatment comprised the second group. The third group included genes with down-regulated expression in the 3 h NaHCO_3_ treatment and up-regulated expression in the 6 h NaHCO_3_ treatment. Genes with down-regulated expression in the 3 h and 6 h NaHCO_3_ treatments comprised the fourth group. The fifth group included genes with up-regulated expression in the 3 h and 6 h NaHCO_3_ treatments.

### 2.5. SNP Analysis of RNA-Seq Data

A total of 578,863 single nucleotide polymorphism (SNP) loci were obtained from the RNA-seq data mapped against the reference genome. A-to-G transitions were the most frequent, accounting for 30.77% of the total. C-to-G transversions accounted for only 7.39% of the SNP mutation types (Figure 5A). There were 47,833, 60,567, and 64,727 SNPs located in the introns, upstream regions, and downstream regions of genes. Additionally, 31,196 synonymous coding and 27,791 nonsynonymous coding SNPs were detected (Figure 5B).

We next analyzed the number of SNP loci, the ratio of transition types, the ratio of transversion types, and the ratio of heterozygous SNP loci identified from NaHCO_3_ treatments for 3 h and 6 h. The total number of SNP loci ranged from 262,684 to 275,847; the total number of SNP loci in genic regions ranged from 231,583 to 241,463; and the total number of SNP loci in the intergenic regions ranged from 31,101 to 34,383. The ratio of transition types was 61.3%; the ratio of transversion types was 38.7%; and the ratio of heterozygous SNP loci was approximately 61% (Table 2).

### 2.6. Confirmation of RNA-Seq Data Using qRT-PCR Analysis

To further evaluate the validity of the RNA-seq data, quantitative real-time reverse-transcription PCR (qRT-PCR) analysis was conducted. Twelve DEGs were randomly selected for qRT-PCR. Our results revealed that the relative expression levels of these genes were consistent with the fragments per kilobase of exon per million fragments mapped (FPKM) values (Figure 6A), and their correlation coefficient (R^2^) was 0.97, demonstrating that the expression of these genes based on qRT-PCR was strongly positively related to the results of the RNA-seq analysis (Figure 6B). Thus, the differential expression analysis revealed that the RNA-seq data were reliable.

### 2.7. Oxidative Stress and Cell Membrane Damage Were Alleviated in Plants Overexpressing WRKY70 and NAC9

The expression of the *WRKY70* and *NAC9* TF genes was significantly up-regulated under NaHCO_3_ treatment compared with the control. Therefore, these two genes were subjected to additional analyses. Birch plants overexpressing *WRKY70* and *NAC9* were obtained using transient *Agrobacterium*-mediated transformation. qRT-PCR was conducted to analyze the expression levels of *WRKY70* and *NAC9.* The relative expression of *WRKY70* and *NAC9* was significantly higher in *WRKY70*- and *NAC9*-overexpressing plants than in pROKII-35S plants (Figure 7). The results indicated that this transient transformation system is efficient for the transformation of genes. In addition, the relative expression of *WRKY70* and *NAC9* was approximately 300 times higher in *WRKY70*- and *NAC9*-overexpressing plants than in pROKII-35S plants under saline–alkali stress, suggesting that the expression of *WRKY70* and *NAC9* can be strongly induced by stress.

Furthermore, 3, 3′-diaminobenzidine (DAB) and nitroblue tetrazolium (NBT) staining was used to detect H_2_O_2_ and O^2−^ levels to determine whether saline–alkali tolerance was strengthened in *WRKY70*- and *NAC9*-overexpressing plants. Leaves from *WRKY70*- and *NAC9*-overexpressing plants and pROKII-35S plants were stained with DAB or NBT; the stained leaves from water-treated plants were used as controls. Under saline–alkali treatment, the H_2_O_2_ and O^2−^ levels in the leaves of *WRKY70*- and *NAC9*-overexpressing plants were greatly reduced compared with those in pROKII-35S plants (Figure 8). The content of H_2_O_2_ and O^2−^ is negatively related to the reactive oxygen species (ROS) scavenging ability of plants. Therefore, our results suggested that the ROS scavenging ability of *WRKY70*- and *NAC9*-overexpressing plants was enhanced. In addition, Evans blue staining was used to detect cell membrane damage. Less intense blue staining of *WRKY70*- and *NAC9*-overexpressing plants was observed, compared with that of pROKII-35S plants under saline–alkali treatment (Figure 8), indicating that cell death was reduced.

### 2.8. Physiological Characterization of WRKY70- and NAC9-Overexpressing Plants

In this study, superoxide dismutase (SOD) and peroxidase (POD) activities, H_2_O_2_ content, and electrolyte leakage were used to assess the resistance of *WRKY70*- and *NAC9*-overexpressing plants to saline–alkali stress, as well as that of the pROKII-35S transformants (Figure 9). Water treatment was used as a control. SOD and POD play important roles in the removal of ROS in plants under stress. Our results indicated that SOD and POD activities in *WRKY70*- and *NAC9*-overexpressing plants were significantly higher in *WRKY70*- and *NAC9*-overexpressing plants than in pROKII-35S plants under saline–alkali stress (Figure 9A,B). The H_2_O_2_ content was substantially down-regulated in *WRKY70*- and *NAC9*-overexpressing plants, relative to that in pROKII-35S plants (Figure 9C). Cell death was evaluated using the electrolyte leakage rate. Electrolyte leakage was lower in *WRKY70*- and *NAC9*-overexpressing plants than in pROKII-35S plants under saline–alkali stress (Figure 9D). These results indicate that *WRKY70* and *NAC9* could enhance the ROS scavenging ability and inhibit cell death in plants.

### 2.9. Expression Changes of Target Genes in WRKY70- and NAC9-Overexpressing Plants

To further analyze whether *WRKY70* and *NAC9* could affect the expression of target genes, the relative expression levels of *SOD* and *POD* in *WRKY70*- and *NAC9*-overexpressing plants and pROKII-35S plants under saline–alkali stress were examined. Under NaHCO_3_ treatment, the relative expression levels of *SOD* and *POD* were higher in *WRKY70*- and *NAC9*-overexpressing plants, relative to the control, than in pROKII-35S plants (Figure 10). The results indicated that *WRKY70* and *NAC9* can regulate the expression of *SOD* and *POD*.

## 3. Discussion

The main aim of this study was to achieve an improved understanding of the molecular mechanism of saline–alkali tolerance and identify some key genes and regulatory pathways that play key roles in the response to saline–alkali stress in *B. platyphylla*. Thus, we performed high-throughput sequencing and comparative RNA-seq analysis of *B. platyphylla* roots under saline–alkali stress (0.2 M NaHCO_3_).

Plants activate a series of complex regulatory processes to mediate adaptation to various types of stresses. In our study, some GO terms related to cellular response to hypoxia were identified; for example, *CYP82C2* DEG was significantly up-regulated under NaHCO_3_ treatments for 3 h and 6 h compared with the control. A previous study has shown that *CYP82*, a member of the cytochrome P450 family, is involved in the resistance of plants to biotic and abiotic stress [15]. The results of our study indicate that *CYP82C2* plays a positive regulatory role in the root system of birch plants experiencing alkali-stress-induced hypoxia damage.

Generally, secondary metabolites play a role in the defense response to stress [16]. In this study, some DEGs that are potentially involved in secondary metabolic processes were identified. For example, three CADs involved in lignin biosynthesis were identified (*CAD1*, *CAD3*, and *CAD5*). The expression of these three genes was mainly down-regulated. A previous study has shown that the expression of *CAD* genes is generally down-regulated under salt stress [8]. Our results are consistent with the results of this previous study.

TFs play key roles in regulatory cascades and are essential for the regulation of gene expression; they also play important roles in regulating the acclimation response of plants to various types of stress. We identified some TF families from the RNA-seq data of birch, such as DREB, WRKY, NAC, ERF, and bHLH TFs, involved in stress, secondary metabolic processes, and the jasmonic acid signaling pathway. A previous study has shown that the expression of a *DREB* gene under a stress-inducible promoter enhanced the drought and frost tolerance of transgenic barley and the frost tolerance of transgenic wheat seedlings [17]. Another study has shown that the overexpression of *RcDREB2B* from *Rosa chinensis* enhances sensitivity to high salt concentrations, abscisic acid, and polyethylene glycol at the germination and post-germination stages in *Arabidopsis*, suggesting that *RcDREB2B* negatively regulates the abiotic stress resistance of plants [18]. In our study, no differences in the expression of genes encoding three DREB TFs in the 3 h NaHCO_3_ treatment group and control were observed; however, the expression of these genes was down-regulated in the 6 h NaHCO_3_ treatment, relative to the control. Thus, the results of our study suggest that DREB TFs are involved in complex regulatory pathways. Previous studies have shown that WRKY TFs can negatively or positively regulate abiotic stress tolerance in plants [19,20,21]. In our study, the expression of *WRKY41* and *WRKY46* was up-regulated, but the up-regulation of the expression of these genes was not pronounced under NaHCO_3_ treatment for 3 h; however, their expression was significantly down-regulated in the 6 h NaHCO_3_ treatment. The expression of *WRKY70* was significantly up-regulated under NaHCO_3_ treatment for 3 h, but no significant differences in the expression of this gene were observed in the 6 h NaHCO_3_ treatment. Thus, the three WRKY TFs might play multiple roles in the response to saline–alkali stress in birch plants. Some studies have shown that NAC, ERF, and bHLH TFs can respond to various types of abiotic stress in plants [22,23,24,25,26]. In our study, the expression of *NAC9*, *ERF019*, and *bHLH162* was up-regulated in the 3 h NaHCO_3_ treatment but down-regulated in the 6 h NaHCO_3_ treatment, relative to the control. However, the expression of *ERF012* was down-regulated in both the 3 h and 6 h NaHCO_3_ treatments. These findings indicate that NAC, ERF, and bHLH TFs might play diverse roles in the stress responses of birch plants.

In our study, the expression of *WRKY70* and *NAC9* was substantially up-regulated in the 3 h NaHCO_3_ treatment, relative to the control. Therefore, birch plants overexpressing these two genes were used to study their stress resistance. Plants are often exposed to various adverse conditions that can lead to the accumulation of ROS [27]. Therefore, ROS scavenging is important for the resistance of plants to various types of stress. Two major ROS species, H_2_O_2_ and O^2−^, are important molecules in cells that are involved in oxidative injuries and stress signaling [28]. In our study, DAB and NBT staining revealed that ROS accumulation was lower in *WRKY70-* and *NAC9*-overexpressing plants than in pROKII-35S plants under saline–alkali treatment (Figure 7), and this was consistent with the observed H_2_O_2_ levels (Figure 8C). SOD and POD played key roles in the removal of ROS. SOD and POD activities were significantly higher in *WRKY70*- and *NAC9*-overexpressing plants than in pROKII-35S plants under saline–alkali stress (Figure 8A,B). Given that SOD and POD activities in *WRKY70*- and *NAC9*-overexpressing plants were altered in response to saline–alkali stress, we measured the expression of *SOD* and *POD*. Under NaHCO_3_ treatment, the expression of all *SOD* and *POD* genes was higher in *WRKY70*- and *NAC9*-overexpressing plants than in pROKII-35S plants (Figure 9). These findings suggest that *WRKY70* and *NAC9* enhance SOD and POD activities by altering the expression of *SOD* and *POD* genes. Overall, these results indicate that *WRKY70* and *NAC9* could enhance SOD and POD activities by regulating the expression of *SOD* and *POD* to enhance ROS scavenging. The results of the Evans blue staining (Figure 7) were consistent with the electrolyte leakage rates (Figure 8D). The *WRKY70*- and *NAC9*-overexpressing plants had significantly less intense Evans blue stain, indicating that electrolyte leakage rates were lower in *WRKY70*- and *NAC9*-overexpressing plants compared with pROKII-35S plants. These findings suggest that *WRKY70* and *NAC9* reduced cell death to enhance resistance to saline–alkali stress in plants.

## 4. Materials and Methods

### 4.1. Plant Growth and Stress Treatments

Birch seeds were sown in plastic pots with a mixture of perlite/vermiculite/soil (1:1:3) and subjected to a 16 h/8 h light/dark cycle, at an average temperature of 25 °C, and 60–70% relative humidity in a greenhouse. The pots were watered twice daily until seed germination, after which they were watered once daily.

After two months of cultivation, healthy birch seedlings approximately 20 cm in height were treated with 0.2 M NaHCO_3_ for 3 h and 6 h. Specifically, the 3 h treatment was performed backwards, with 6 h being the final treatment time point. The control plants were treated with fresh water for 6 h. The roots of birch seedlings were collected. Three independent biological replicates were conducted, and each replicate comprised six seedlings. All samples were quickly frozen in liquid nitrogen and stored at −80 °C.

### 4.2. RNA Extraction, Library Construction, and RNA-Seq

RNA was extracted using the RNAprep Pure Plant Kit (Tiangen, Beijing, China) per the manufacturer’s instructions. The concentration and purity of RNA were measured using a NanoDrop2000 spectrophotometer (Thermo Fisher Scientific, Wilmington, NC, USA). The integrity of RNA was evaluated using the RNA Nano 6000 Assay Kit of the Agilent Bioanalyzer 2100 system (Agilent Technologies, Santa Clara, CA, USA).

The cDNA library was constructed per the manufacturer’s instructions of the NEB Next Ultra RNA Library Prep Kit for Illumina (NEB, E7530) and NEB Next Multiplex Oligos for Illumina (NEB, E7500). Briefly, the enriched mRNA was fragmented into RNA inserts that were approximately 200 nt, which were used to synthesize first-strand cDNA and second-strand cDNA. The double-stranded cDNA was end-repaired, A-tailed, and ligated to adaptors. The appropriate fragments were isolated using Agencourt AMPure XP beads (Beckman Coulter, Brea, CA, USA) and enriched via PCR amplification. Finally, the constructed cDNA library was sequenced on a flow cell using an Illumina HiSeq™ sequencing platform.

### 4.3. RNA-Seq Analysis Using Reference Genome-Based Read Mapping

Low-quality reads, such as reads containing adaptors, reads with greater than 5% unknown nucleotides, or reads with Q20 values less than 20% (percentage of sequences with sequencing error rates less than 1%), were removed by a Perl script. The clean reads that were filtered from the raw reads were mapped to the *B. platyphylla* genome using Hisat2 tools. Duplicate sequences were removed from the aligned sequences in BAM/SAM format. Gene expression levels were measured using FPKM values.

### 4.4. Analysis of DEGs

Analysis of DEGs in *B. platyphylla* plants under control conditions and after 3 h and 6 h of NaHCO_3_ treatment was conducted using DESeq2 [29]. The *p*-value threshold in multiple tests for determining the significance of differences was calculated using the false discovery rate (FDR) control method. Here, only genes with an absolute value of log_2_ ratio ≥ 2 and a FDR significance score < 0.01 were used for subsequent analysis.

### 4.5. Gene Functional Annotation

Genes were compared, via BLASTX, against various protein databases, including the National Center for Biotechnology Information (NCBI) non-redundant protein (Nr) database, Kyoto Encyclopedia of Genes and Genomes, COG, and Swiss-Prot database with an E-value cutoff of 10^−5^. In addition, searches of genes were conducted against the NCBI non-redundant nucleotide sequence database, using BLASTn, with an E-value cutoff of 10^−5^. Genes were retrieved according to the best BLAST hit (highest score), along with their protein functional annotations.

Gene Ontology (GO) annotations were obtained using the Nr BLAST results and the Blast2GO program [30]. All the annotated genes were mapped to the GO terms in the database, and the number of genes associated with each term was determined. A Perl script was used to plot the GO functional classification of the unigenes with their GO terms to visualize the distribution of gene functions. The obtained annotations were enriched and refined using TopGo (R package). The gene sequences were also aligned to the COG database to predict and classify functions.

### 4.6. SNP Analysis

SNPs are genetic markers caused by single nucleotide variation in the genome. SNP calling Picard tools v1.41 and SAMtools v0.1.18 were used to sort and remove duplicated reads, and merge the BAM alignment results of each sample. GATK2 software was used to identify single-base mismatches between the sequenced samples and the reference genome to identify potential SNP loci. Raw vcf files were filtered using the GATK standard filter method and other parameters (cluster Window Size: 10; MQ0 >= 4 and (MQ0/(1.0×DP)) > 0.1; QUAL < 10; QUAL < 30.0 or QD < 5.0 or HRun > 5), and only SNPs with distances less than 5 were retained.

### 4.7. Quantitative Real-Time PCR (qRT-PCR) Verification

Total RNA was reverse-transcribed into cDNA using a PrimeScript RT Master Mix kit (Takara Bio Inc. Kusatsu, Shiga, Japan). A total of 12 previously identified DEGs were randomly selected for qRT-PCR verification. The genes encoding ubiquitin (GenBank accession number: FG065618) and tubulin (GenBank accession number: FG067376) were used as the internal controls; all primer sequences are listed in Appendix A. PCR reactions were performed in a volume of 20 μL with 10 μL of TB Green Premix Ex Taq; 2 μL of 0.5 μg/μL cDNA; 1 μL of 10 μM Primer-F; and 1 μL of 10 μM Primer-R. The thermal cycling conditions were as follows: 95 °C for 30 s; 95 °C for 20 s, 55 °C for 30 s, and 72 °C for 30 s for 44 cycles; and 60 °C for 15 s. The relative expression levels of the genes were calculated using the delta–delta CT method [31] to verify the results of the RNA-seq analysis. Three independent experiments were conducted.

### 4.8. Plasmid Construct Containing TF Genes and Plant Transformation

TFs can control the expression of various target genes and therefore play important roles in stress responses in plants. In this study, some genes encoding TFs were differentially expressed according to the RNA-seq data. We used two up-regulated DEGs encoding the TFs WRKY70 (GenBank accession number: OR052244) and NAC9 (GenBank accession number: OR052245) for further analysis.

The cDNA sequences of *WRKY70* and *NAC9* were obtained from the birch genome. The open reading frames of *WRKY70* and *NAC9* were cloned using In-Fusion ligase (Novoprotein, Suzhou, China) into the pROKII vector digested with Sma I (Takara Bio Inc. Kusatsu, Shiga, Japan) under control of the CaMV 35S promoter. The primers and amplicon sizes are shown in Appendix A. The pROKII-35S::gene construct and pROKII-35S empty vector were transformed into *Agrobacterium* EHA105 competent cells, and then into one-month-old birch plants, via the *A. tumefaciens*-mediated transient transformation method following the method described in a previous study [32].

### 4.9. Relative Expression and Stress Tolerance Analysis of WRKY70 and NAC9-Overexpressing Plants

Birch plants overexpressing *WRKY70* and *NAC9* were treated with 0.2 M NaHCO_3_ for 6 h. The pROKII-35S empty vector transformants were also treated with NaHCO_3_. Water treatment was used as a control.

Total RNA from each sample was reverse-transcribed into cDNA. The relative expression levels of *WRKY70* and *NAC9* in plants overexpressing *WRKY70* and *NAC9* and pROKII-35S birch plants were determined under control and 0.2 M NaHCO_3_ conditions using qRT-PCR. The primer sequences are listed in Appendix A. The relative expression levels of the genes were determined using the 2^−ΔΔCt^ [31]. Three independent experiments were conducted.

The detached leaves of birch plants were infiltrated with DAB (1.0 mg/mL) and NBT (0.5 mg/mL) according to a previously described method [28]. Evans blue (1.0 mg/mL) staining was performed to detect cell death following previously described procedures [33]. SOD and POD activities, H_2_O_2_ content, and electrolyte leakage were measured following previously described procedures [34,35]. Three independent biological replicates were conducted.

Total RNA of birch plants overexpressing *WRKY70* and *NAC9* and pROKII-35S birch plants was extracted and treated with DNase I before being reverse-transcribed into cDNA using a PrimeScript RT reagent Kit (Lab). Real-time PCR was conducted on six *SOD* and eight *POD* genes. The primer sequences are listed in Appendix A.

### 4.10. Statistical Analysis

All statistical analyses were conducted using SPSS 22.0 software (IBM, Bloomington, IL, USA). Analysis of variance was used to determine the significance of differences between groups; the threshold for statistical significance was *p* < 0.05.

## 5. Conclusions

In sum, 595 and 607 alkali stress-responsive DEGs were identified in the roots of *B. platyphylla* following exposure to saline–alkali stress (NaHCO_3_ treatment) for 3 h and 6 h, respectively. These genes were involved in stress, signal transduction, secondary metabolic process, regulation of jasmonic acid, abiotic stimulus signaling pathway, and TFs. The SNP loci in the DEGs were associated with alkali tolerance in birch. In addition, the overexpression of *WRKY70* and *NAC9* increased alkali stress tolerance in plants, suggesting that *WRKY70* and *NAC9* TFs play key roles in mediating resistance to saline–alkali stress in birch plants. The results of this study enhance our understanding of the stress tolerance of *B. platyphylla* at the molecular level, and provide new insights that have implications for studies of the saline–alkali stress tolerance of other plants. The two key genes *WRKY70* and *NAC9* could also be used in plant molecular breeding programs to generate germplasm resources with high saline–alkali tolerance in the future.

## Figures and Tables

**Figure 1 plants-12-02435-f001:**
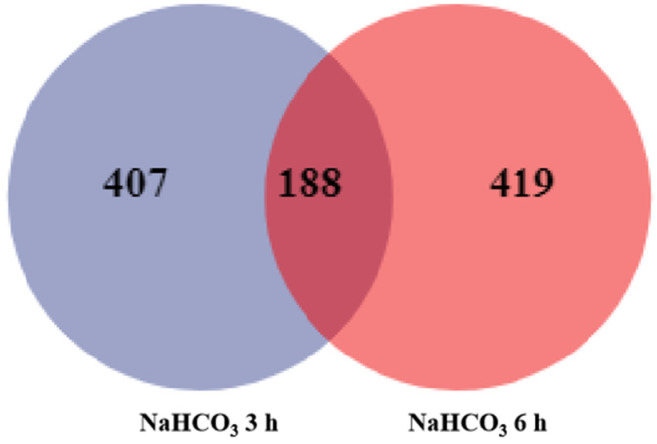
The overlap of DEGs according to RNA-seq analysis. All expressed genes in NaHCO_3_-treated plants at 3 h and 6 h. Significance was determined as FDR < 0.01.

**Figure 2 plants-12-02435-f002:**
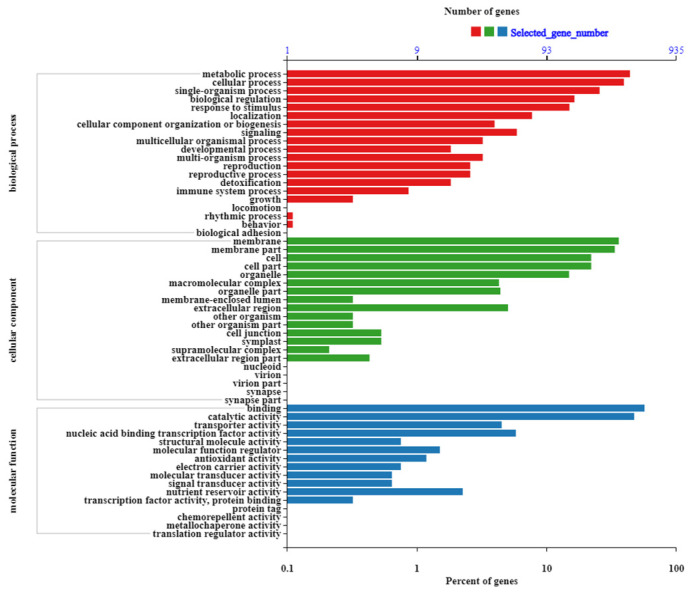
GO annotations assigned to DEGs according to RNA-seq analysis. The abscissa shows the GO terms; the left ordinate is the percentage of the number of genes; and the right ordinate is the number of genes.

**Figure 3 plants-12-02435-f003:**
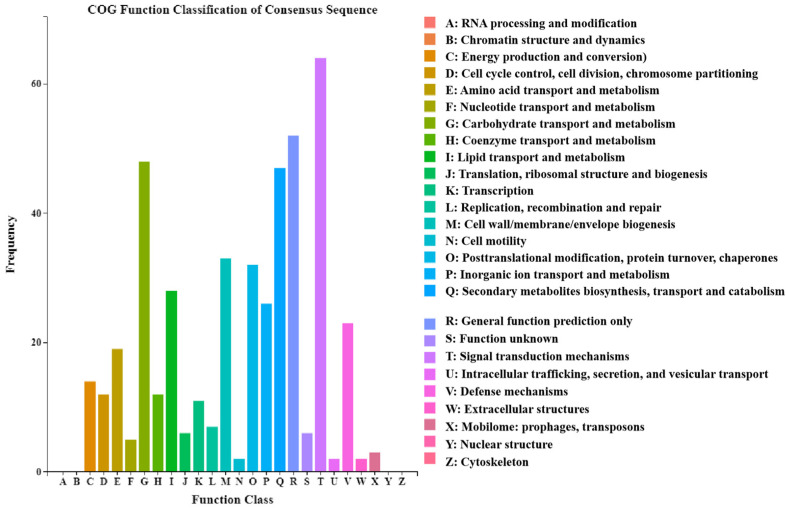
COG annotations assigned to DEGs according to the RNA-seq analysis. The abscissa is the content of each COG classification, and the ordinate is the number of genes.

**Figure 4 plants-12-02435-f004:**
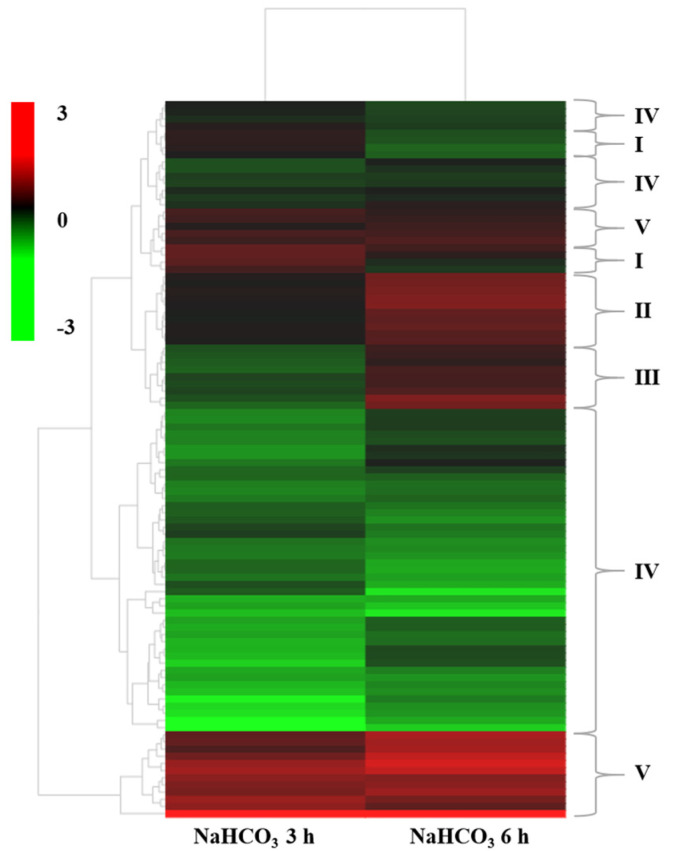
Hierarchical clustering analysis of DEGs according to the RNA-seq data. Heatmap analysis of 100 DEGs in the control, and plants treated with 0.2 M NaHCO_3_ for 3 h and 6 h. I–V: the first group–the fifth group. Red and green in the heatmap indicate up-regulated and down-regulated genes, respectively.

**Figure 5 plants-12-02435-f005:**
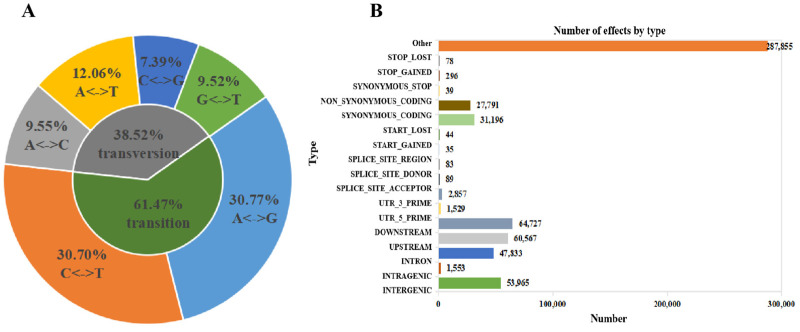
Analysis of SNPs according to the RNA-seq analysis. (**A**) Statistics of SNP mutation types. (**B**) SNP annotation classification. The vertical axis is the region or type of SNP, and the horizontal axis is the number of categories.

**Figure 6 plants-12-02435-f006:**
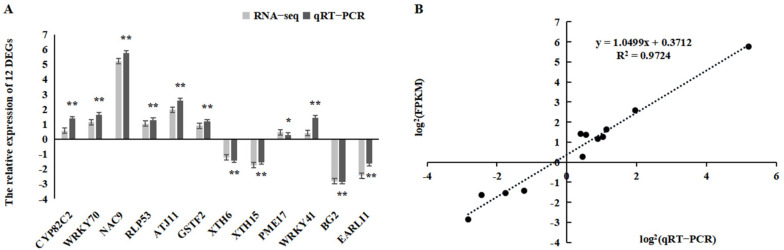
Comparison of the relative abundance of 12 selected transcripts as determined by qRT-PCR and FPKM values for RNA-seq. (**A**) Relative expression levels of the 12 selected transcripts. (**B**) Correlation between relative abundance values. Error bars were obtained from multiple qRT-PCR replicates. * indicates a significant difference (*p* < 0.05). ** indicates a significant difference (*p* < 0.01).

**Figure 7 plants-12-02435-f007:**
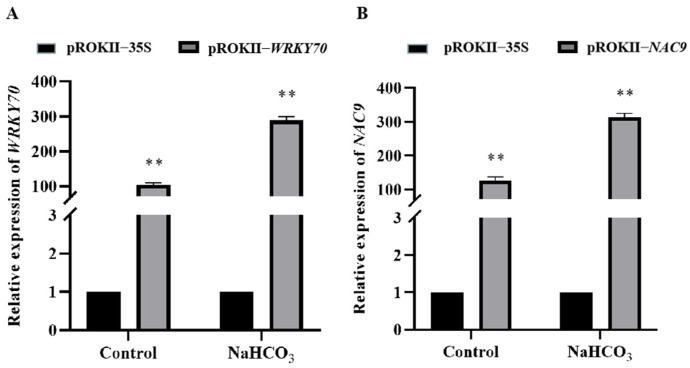
The relative expression levels of *WRKY70* and *NAC9* in *WRKY70*- and *NAC9*-overexpressing plants. Error bars were obtained from multiple qRT-PCR replicates. ** indicates a significant difference (*p* < 0.01).

**Figure 8 plants-12-02435-f008:**
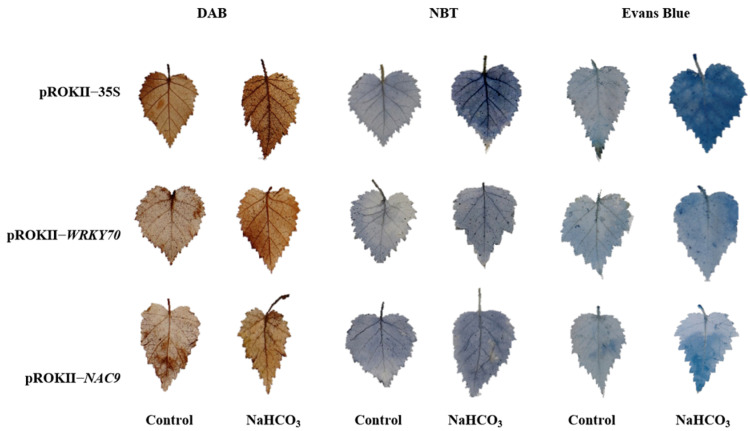
Analysis of ROS accumulation and cell-membrane damage among transgenic and control birch plants. Birch plants were transiently transformed with 35S: *WRKY70* and *NAC9* to generate *WRK70*- and *NAC9*-overexpressing plants, and empty pROKII to provide plants as the control. After 6 h of treatment with 0.2 M NaHCO_3_, *WRK70*- and *NAC9*-overexpressing plants and pROKII-35S plants were individually stained with DAB to visualize H_2_O_2_ levels, NBT to visualize O^2−^, and Evans blue to visualize cell membrane damage.

**Figure 9 plants-12-02435-f009:**
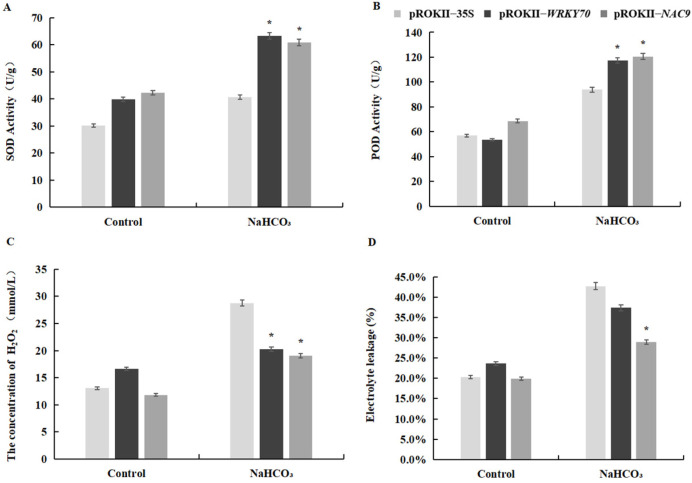
Physiological analysis of transgenic *WRKY70*- and *NAC9*-overexpressing plants and control plants. SOD and POD activities, H_2_O_2_ content, and electrolyte leakage in pROKII-35S and transgenic plants under saline–alkali treatment. (**A**) Measurement of SOD activity, (**B**) POD activity, (**C**) H_2_O_2_ content, and (**D**) electrolyte leakage. The error bars indicate the standard deviation of three biological replicates. * indicates a significant difference (*p* < 0.05).

**Figure 10 plants-12-02435-f010:**
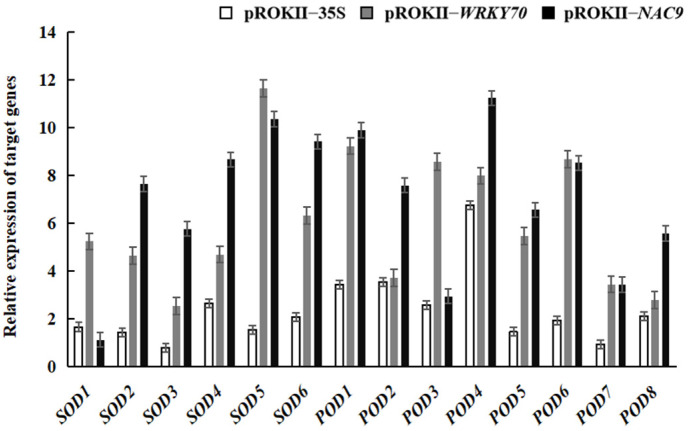
Relative expression analyses of target genes in *WRKY70* and *NAC9* transgenic birch plants. The error bars indicate the standard deviation of three biological replicates.

**Table 1 plants-12-02435-t001:** The characteristics of the DEGs in *B. platyphylla* under different lengths of NaHCO_3_ exposure.

Items	Number of Genes	
	Control versus 3 h	Control versus 6 h
Total	595	607
Up-regulated	290	199
Down-regulated	305	408

**Table 2 plants-12-02435-t002:** SNP data statistics from the RNA-seq analysis.

Names	SNP Number	Genic SNP	Intergenic SNP	Transition	Transversion	Heterozygosity
Control	277,460	242,456	35,005	61.32%	38.68%	61.71%
NaHCO_3_ 3 h	275,847	241,463	34,383	61.30%	38.70%	61.55%
NaHCO_3_ 6 h	262,684	231,583	31,101	61.30%	38.70%	61.80%

## Data Availability

All data relevant to the main findings of this study are included within the article.

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
