# Peer review of "Exploration of the Regulatory Pathways and Key Genes Involved in the Response to Saline–Alkali Stress in Betula platyphylla via RNA-Seq Analysis"

_plants, 2023, doi:10.3390/plants12132435_

Round 1

Reviewer 1 Report

This study is useful for understanding the genetic mechanisms regulating the tolerance of Betula platyphylla to stress induced by saline-alkali soil.

Overall the article is well structured and the results are clearly presented.

The methodology is appropriate.

The citations of the scientific literature have to be improved: some references in both the Introduction and the Discussion are missing (I added just one relevant reference, other references must be added by the Authors).

The English style can be improved further (see text editings as notes in the text: attached file)

Author Response

The citations of the scientific literature have to be improved: some references in both the Introduction and the Discussion are missing (I added just one relevant reference, other references must be added by the Authors).

Response: Sorry, we accidentally deleted several references, and now we have added them in the whole manuscript.

Reviewer 2 Report

This research is well designed and is interested in the defense against alkaline-saline stress for birch and plants in general. Results are well presented and conclusions based on the results are appropriate.

My major comments and suggestions are related to language and grammar, as listed here:

Overall, all abbreviations should be written with full words when first time mentioned

Line 23 Fufther

Line 31 „Saline-alkali soils exceeds 10 × 108 hectares of global land surface“, is this corectly written? In the reference (1) it says: „10% of the total arable land as being affected by salinity and sodicity, one billion hectares are covered with saline and/or sodic soils, and between 25% and 30% of irrigated lands are salt-affected and essentially commercially unproductive, global distribution of salt-affected soils are 954 million ha, FAO in 1988 presented 932 million ha salt-affected soils, of almost 1500 million ha of dryland agriculture, 32 million ha are salt-affected.“

Line 32 Na+, plus should be in superscript

Line 81 instead „were down-regulated for 6 h“, it would be better to write „were down-regulated after 6 h of treatment“

Line 82 instead „with different 82 time NaHCO3“, it should be written „with NaHCO3 for different period of time“

Line 86 not clear what was the point „All expressed genes set for NaHCO3 treatment plants at 3 and 6 h“

Figure 2 at x-axis, the names of the processes are hardly readable. Maybe you could write an abreviation (letter, number) and next to the figure the list of abbreviations in bigger font size (the same like in Figure 3)

Line 117 „The genes that slightly up-regulated“, should be „The genes that were slightly up-regulated“

Line 117 „during stress period“, which stress period, 6 h?

Line 118 „The third group includes the genes that down-regulated at stress for 3 h, and up-regulated at stress for 6 h“; that ARE down-regulated

In general, you should uniform naming of the groups; e.g. stress for 3 h / treatment for 3 h / at 3 h / at 3 h time point…..

Line 124 NaHCO3; number 3 should be in subscript

Line 128 „were most frequent“, THE is missing before „most“

Line 147 „The expression of 12 DEGs that were randomly selected.“, the verb is missing

Line 179 NaHCO3; number 3 should be in subscript

Line 180 H2O2; numbers 2 should be in subscript and 2- in O2− should be in superscript

Figure 8 legend WRKY70- and NAC9, name of genes in Italic; H2O2; numbers 2 should be in subscript

Line 212 WRKY70- and NAC9, name of genes in Italic

Discussion

Genes should be written in Italics

Line 241 name of the species in Italics (Rosa chinensis)

Line 243 same as before (Arabidopsis)

Line 265 „H2O2 and O2− are important molecules in cells“, O2- is not a molecule

Line 269 H2O2; numbers 2 should be in subscript

Line 314, 320 name of the species in Italics

Line 330 and 332 10-5 (-5 in superscript)

Chapters 4.8 and 4.9 Genes and names of the species should be written in Italics

Line 400 I couldn't find any file at this web page (Supplementary Materials: The following supporting information can be downloaded at: www.mdpi.com/xxx/s1, Figure S1: title; Table S1: title; Video S1: title.)

Author Response

Comments and Suggestions for Authors

  • Overall, all abbreviations should be written with full words when first time mentioned

Response: Thank you for your suggestion, and we have finished the revision in the whole manuscript.

  • Line 23 Fufther

Response: Sorry, due to our negligence, there was an error in the spelling of the word, and we replaced it with” Additional”(line 23).

  • Line 31 „Saline-alkali soils exceeds 10 × 108 hectares of global land surface“, is this corectly written? In the reference (1) it says: „10% of the total arable land as being affected by salinity and sodicity, one billion hectares are covered with saline and/or sodicsoils, and between 25% and 30% of irrigated lands are salt-affected and essentially commercially unproductive, global distribution of salt-affected soils are 954 million ha, FAO in 1988 presented 932 million ha salt-affected soils, of almost 1500 million ha of dryland agriculture, 32 million ha are salt-affected.“

Response: Sorry, due to our negligence, there was an error, we have revised the manuscript (line 32-33).

  • Line 32 Na+, plus should be in superscript

Response: Sorry, due to our negligence, there was an error, we have revised it in the manuscript (line 35). 

  • Line 81 instead „were down-regulated for 6 h“, it would be better to write „were down-regulated after 6 h of treatment“

Line 82 instead „with different 82 time NaHCO3“, it should be written „with NaHCO3 for different period of time“

Response: Thank you for your suggestion, I have finished the modification in the manuscript (line 82-90).

  • Line 86 not clear what was the point „All expressed genes set for NaHCO3 treatment plants at 3 and 6 h“

Response: Thank you for your suggestion, I have finished the modification in the manuscript (line 92).

  • Figure 2 at x-axis, the names of the processes are hardly readable. Maybe you could write an abreviation (letter, number) and next to the figure the list of abbreviations in bigger font size (the same like in Figure 3)

Response: Thank you for your suggestion, I have finished the modification in Figure 2.

  • Line 117 „The genes that slightly up-regulated“, should be „The genes that were slightly up-regulated“

Line 117 „during stress period“, which stress period, 6 h?

Line 118 „The third group includes the genes that down-regulated at stress for 3 h, and up-regulated at stress for 6 h“; that ARE down-regulated

In general, you should uniform naming of the groups; e.g. stress for 3 h / treatment for 3 h / at 3 h / at 3 h time point…..

Response: We have revised the paragraph in the manuscript (line 117-132).  

  • Line 124 NaHCO3; number 3 should be in subscript.

Line 128 „were most frequent“, THE is missing before „most“

Line 179 NaHCO3; number 3 should be in subscript

Line 180 H2O2; numbers 2 should be in subscript and 2- in O2− should be in superscript

Line 180 H2O2; numbers 2 should be in subscript and 2- in O2− should be in superscript

Line 212 WRKY70- and NAC9, name of genes in Italic

Response: We have finished the modification in the manuscript.

  • Line 147 „The expression of 12 DEGs that were randomly selected.“, the verb is missing

Response: Sorry, we revised it in the manuscript (line 159).  

  • Discussion

Genes should be written in Italics

Line 241 name of the species in Italics (Rosa chinensis)

Line 243 same as before (Arabidopsis)

Line 265 „H2O2 and O2− are important molecules in cells“, O2- is not a molecule

Line 269 H2O2; numbers 2 should be in subscript

Line 314, 320 name of the species in Italics

Line 330 and 332 10-5 (-5 in superscript)

Chapters 4.8 and 4.9 Genes and names of the species should be written in Italics

 Response: Thank you for your suggestion, I have finished the modification in the manuscript.

  • Line 400 I couldn't find any file at this web page (Supplementary Materials: The following supporting information can be downloaded at: www.mdpi.com/xxx/s1, Figure S1: title; Table S1: title; Video S1: title.)

Response: Thank you for your comments. We have added the information in the manuscript (line 452-457).

Reviewer 3 Report

The article makes no sense, is written very carelessly, there are many language errors and typos

In the manuscript you mention transcriptomic analysis and RNA-seq. Were these two different analyses? If not, I don’t see the reason of using both words.

Is is also not clear to me how did you detect “47,833, 60,567 and 64,727 SNPs located in the introns, upstream and downstream regions of genes” and “the total number of SNP loci in the intergenic regions” if you studied a transcriptome? Do you mean the UTRs by “upstream and downstream regions of genes”? Why are there intergenic regions in your transcriptome if you used a polyA tail? In this regard, you should investigate, what was the ratio of RNA types in your transcript.

“healthy birch seedlings approximately 20 cm in height with similar growth conditions were treated with 0.2 M NaHCO 3 for 3 h and 6 h.”

Why did you choose this concentration and these time points?

“The open reading frame (ORF) of WRKY70 and NAC9 was cloned into the

pROKII vector under the control of the CaMV 35S promoter, respectively (The primers

and amplicon sizes are shown in Table S5). The pROKII-35S::gene construct and pROKII-

35S empty vector was separately transformed”

What was the vector you used how exactly did you clone the gene (what enzymes did you use, etc.)?

“transient transformation method according to previous report [29]”

Why did you choose a transient transformation? How did you prove that plants were transformed? In the original method [29] plants were just soaked in agrobacterium suspension and “After culturing for 48 h, the plants were assumed to have been transformed”. There should be proofs that transient expression of the genes really occur.

“The birch plants overexpressing WRKY70 and NAC9 were treated with 0.2 M Na-

HCO 3 for 6 h.”

You do not describe how did you prove that these plants were overexpressing these genes. Also describe the scheme of treatment.

“The detached leaves of birch plants were infiltrated with 3’-diaminobenzidine (DAB,

1.0 mg/mL) and nitroblue tetrazolium (NBT, 0.5 mg/mL) based on published method.”

The reference to the “published method” is absent.

“The DEGs were identified by RNA-seq analysis, the results indicated that there

were 595 and 607 genes significantly differentially expressed after NaHCO 3 stress for 3

and 6 h, respectively (Table 1). Compared with control, 290 DEGs were up-regulated

and 305 DEGs were down-regulated under NaHCO 3 treatment for 3 h, and 199 DEGs

were up-regulated and 408 DEGs were down-regulated for 6 h.”

This part is not well written. It is not clear, what exactly did you compare. Were 595 and 607 genes differentially expressed compared not with the control but with something else? Are 290 and 305 DEGs among these 595 genes? Were genes down-regulated for 6 h (during all this time) or after 6 h treatment? I had to read it several times to figure it out.

“SNP Analysis of RNA-seq Data”

That would be a very interesting analysis if you had a stress-resistant variety. It also doesn’t look like you have found a single mutation in a coding sequence. Why there are non-transcribed sequences (intergenic) in your RNA?

“The expression of 12 DEGs that were randomly selected.”

Why did you select them randomly? The sentence is also unfinished

How exactly did you calculate the gene expression via RT-PCR? It is usually expressed in % of reference gene expression (actin and tubulin in your case). How can you compare this to the number of transcripts from your RNA-seq? Fig 6A looks more like a representation of the expression change but it is not clear what were the start and the end point.

“Plants Overexpressing WRKY70 and NAC9 had Alleviated Oxidative Stress and Cell Mem-

brane Damage”

Why did you choose these genes among “12 DEGs that were randomly selected”? How these random genes appeared to be those most responsive for saline-alkaline stress resistance?

“For example, three cinnamyl-alcohol dehydrogenase (CAD) related to the lig-

nin biosynthesis were found, namely CAD1, CAD3 and CAD5. The expression of these

three genes mainly were down-regulated.”

“In this study, we identified some TF families from the RNA-seq

data of birch, such as DREB, WRKY, NAC, ERF and bHLH TFs involved in stress, second-

ary metabolic processes, and jasmonic acid signaling pathway.”

You do not mention these genes in Results, they just suddenly appear in Discussion

“Previous study indicated that CAD gene was generally down-regulated under salt stress.”

Which study?

You should also indicate the genome you used as a reference, the database you uploaded your own sequences to (such as NCBI) and a reference number.

"saline-alkali" instead of saline-alkaline

a typo in the Title, the Abstract and all over the text

"10 × 108 hectares" “a cut-off E-value of 10-5”

should there be a superscript?

“Illumina(NEB, E7530)and” “Illumina(NEB, E7500).In briefly”

spaces are missing

Author Response

Comments and Suggestions for Authors

  • In the manuscript you mention transcriptomic analysis and RNA-seq. Were these two different analyses? If not, I don’t see the reason of using both words.

Response: Thanks for your suggestion. We revised the word of transcriptomic to RNA-seq in whole manuscript.

  • Is is also not clear to me how did you detect “47,833, 60,567 and 64,727 SNPs located in the introns, upstream and downstream regions of genes” and “the total number of SNP loci in the intergenic regions” if you studied a transcriptome? Do you mean the UTRs by “upstream and downstream regions of genes”? Why are there intergenic regions in your transcriptome if you used a polyA tail? In this regard, you should investigate, what was the ratio of RNA types in your transcript.

“SNP Analysis of RNA-seq Data”That would be a very interesting analysis if you had a stress-resistant variety. It also doesn’t look like you have found a single mutation in a coding sequence. Why there are non-transcribed sequences (intergenic) in your RNA?

Response: In this study, RNA-seq Data were analyzed after comparison with reference genome. Due to incomplete genome annotation information, there are unannotated genes in intergenic regions. Moreover, transcripts containing intron information are identified due to alternative splicing. In addition, RNA-seq can detect the immature mRNA including intron information. 

    In Fig.5, UTR_5_PRIME and UTR_3_PRIME mean the UTR regions of genes. The upstream and downstream regions of genes separately means upstream sequence and downstream sequence of genes (default length: 5000 bp).

We added the content in the part of 4.6. SNP Analysis (Line 379-385).

  • “healthy birch seedlings approximately 20 cm in height with similar growth conditions were treated with 0.2 M NaHCO 3 for 3 h and 6 h.”Why did you choose this concentration and these time points?

Response: We found that 0.2 M NaHCO3 treatment is an appropriate concentration, which can trigger the stress response of birch seedlings, but no wilting or visible damage in birch plants were occurred; therefore this concentration was used for analysis gene expression in birch plants. In addition, we found the more differentially expressed genes (DEGs) in birch plans under short time stress treatment were identified than that under long time stress treatment. So, birch plants were treated by NaHCO3 for 3 h and 6 h, at this study.

  • “The open reading frame (ORF) of WRKY70 and NAC9 was cloned into the pROKII vector under the control of the CaMV 35S promoter, respectively (The primersand amplicon sizes are shown in Table S5). The pROKII-35S::gene construct and pROKII-35S empty vector was separately transformed” What was the vector you used how exactly did you clone the gene (what enzymes did you use, etc.)?

Response: The open reading frame (ORF) of WRKY70 and NAC9 was cloned using In-Fusion ligase into the pROKII vector digested by Sma I, respectively. We added the contents in our manuscript (Line 405-407).

  • “transient transformation method according to previous report [29]”. Why did you choose a transient transformation? How did you prove that plants were transformed? In the original method [29] plants were just soaked in agrobacterium suspension and “After culturing for 48 h, the plants were assumed to have been transformed”. There should be proofs that transient expression of the genes really occur. “The birch plants overexpressing WRKY70 and NAC9 were treated with 0.2 M Na-HCO 3 for 6 h.”You do not describe how did you prove that these plants were overexpressing these genes. Also describe the scheme of treatment.

Response: Transient transformation systems are powerful tools for analyzing the function of genes (Chen et al. 2010). Previous study found that Agrobacterium cells can get into all the tissues of a plant when the plant is soaked in a liquid medium containing Agrobacterium cells for a certain period of time (Ji et al. 2013). In addition, a quick way to determine the stress tolerance of a gene was also developed based on this transient transformation, which allows the accurate analysis of the function of genes involved in stress response in a short time. Therefore, this transiently transformation method may be a powerful tool for characterizing gene function in plant species.

Moreover, we added the experiments of qRT-PCR to detect the relative expression of WRKY70 and NAC9 in overexpressing WRKY70 and NAC9 birch plants based on Agrobacterium tumefaciens-mediated transient transformation method. The part of method (Line 417-442) and results (Line 174-185) was added in our manuscript, and Figure 7 was added.

References:

Chen X, Equi R, Baxter H, Berk K, Han J, Agarwal S, Zale J (2010) A high-throughput transient gene expression system for switchgrass (Panicum virgatum L.) seedlings. Biotechnol Biofuels 3:9

Ji, X., Zheng, L., Liu, Y., Nie, X., Liu, S. and Wang, Y. (2014) A transient transformation system for the functional characterization of genes involved in stress response. Plant Mol. Biol. Rep. 32, 732–739.

  • “The detached leaves of birch plants were infiltrated with 3’-diaminobenzidine (DAB,1.0 mg/mL) and nitroblue tetrazolium (NBT, 0.5 mg/mL) based on published method.”The reference to the “published method” is absent.

Response: Sorry, we accidentally deleted the reference, and now we have added it (Line 424).

  • “The DEGs were identified by RNA-seq analysis, the results indicated that therewere 595 and 607 genes significantly differentially expressed after NaHCO 3 stress for 3 and 6 h, respectively (Table 1). Compared with control, 290 DEGs were up-regulated and 305 DEGs were down-regulated under NaHCO 3 treatment for 3 h, and 199 DEGs were up-regulated and 408 DEGs were down-regulated for 6 h.”

This part is not well written. It is not clear, what exactly did you compare. Were 595 and 607 genes differentially expressed compared not with the control but with something else? Are 290 and 305 DEGs among these 595 genes? Were genes down-regulated for 6 h (during all this time) or after 6 h treatment? I had to read it several times to figure it out.

Response: Sorry, the meaning of this part is a bit ambiguous. So, we revised it in our manuscript (Line 82-90).

  • “The expression of 12 DEGs that were randomly selected.”Why did you select them randomly? The sentence is also unfinished.How exactly did you calculate the gene expression via RT-PCR? It is usually expressed in % of reference gene expression (actin and tubulin in your case). How can you compare this to the number of transcripts from your RNA-seq? Fig 6A looks more like a representation of the expression change but it is not clear what were the start and the end point.Why did you choose these genes among “12 DEGs that were randomly selected”? How these random genes appeared to be those most responsive for saline-alkaline stress resistance?

Response: Sorry, the meaning of this part is a bit ambiguous. So, we revised it in our manuscript (Line 158-165). In addition, in this part, the purpose of the expression analysis of DEGs to verify the RNA-seq data, so random selection of DEGs is reasonable. The qRT-PCR analysis is relative expression of genes using the delta–delta CT method, which can not exactly calculate the gene expression. This confirmation only analyzes the consistent of trend of gene expression between RNA-seq and qRT-PCR, and the accuracy of the data does not affect the results of this experiment. Many studies have used qRT-pCR to verify RNA-seq data (Wang et al. 2013;Guo et al. 2019).

In the part ‘4.4. Differential Gene Expression Analysis’, differential gene with an absolute value of log2 ratio ≥2 and FDR significance score <0.01 were used for subsequent analysis. So, the expressions of DEGs in RNA-seq data were significant.    

Moreover, we added the significance analysis of qRT-PCR in Fig.6A.

References:

Wang C, Gao C, Wang L, Zheng L, Yang C, Wang Y. Comprehensive transcriptional profiling of NaHCO3-stressed Tamarix hispida roots reveals networks of responsive genes. Plant Mol Biol. 2014,84:145–157

Guo H , Zhang C, Wang Y, Zhang Y, Zhang Y, Wang Y, Wang C. Expression profiles of genes regulated by BplMYB46 in Betula platyphylla. Journal of Forestry Research. 2019,30(6):2267-2276.

  • “Plants Overexpressing WRKY70 and NAC9 had Alleviated Oxidative Stress and Cell Membrane Damage”“For example, three cinnamyl-alcohol dehydrogenase (CAD) related to the lignin biosynthesis were found, namely CAD1, CAD3 and CAD5. The expression of these three genes mainly were down-regulated.”“In this study, we identified some TF families from the RNA-seq data of birch, such as DREB, WRKY, NAC, ERF and bHLH TFs involved in stress, secondary metabolic processes, and jasmonic acid signaling pathway.”You do not mention these genes in Results, they just suddenly appear in Discussion.

Response: Thanks for your suggestion. We added the content about these genes in the part of 2.4(Line 117-118).

  • “Previous study indicated that CAD gene was generally down-regulated under salt stress.”Which study?

Response: Sorry, we accidentally deleted the reference, and now we have added it (Line 264).

  • You should also indicate the genome you used as a reference, the database you uploaded your own sequences to (such as NCBI) and a reference number.

Response: Thanks for your suggestion. The cDNA of WRKY70 and NAC9 was cloned  into pROKII vector, and was sequenced, respectively. Now, we have uploaded the sequences of WRKY70 and NAC9 to NCBI and added their Genbank numbers to the part of 4.8 in our manuscript (Line 402-403).

  • Comments on the Quality of English Language"saline-alkali" instead of saline-alkaline a typo in the Title, the Abstract and all over the text

Response: The “saline-alkali” was used in some papers (Zhou et al., 2022; Gao et al., 2022), so we used it.

In addition, the manuscript has been edited by a very experienced editor whose first language is English.

References:

Zhou J,  Qi A,  Wang B,  Zhang X,  Dong Q,  Liu J. Integrated analyses of transcriptome and chlorophyll fluorescence characteristics reveal the mechanism underlying saline–alkali stress tolerance in Kosteletzkya pentacarpos. Front Plant Sci., 2022, 6;13:865572.

Gao Y,  Jin Y,  Guo W,  Xue Y,  Yu L. Metabolic and physiological changes in the roots of two oat cultivars in response to complex saline-alkali stress. Front Plant Sci., 2022, 29;13:835414.

  • "10 × 108 hectares" “a cut-off E-value of 10-5”should there be a superscript? “Illumina(NEB, E7530)and” “Illumina(NEB, E7500).In briefly” spaces are missing.

Response: Sorry, due to our oversight, there are some errors, which we have corrected them.

Round 2

Reviewer 3 Report

Dear Authors!

Your results still don't make sense. Moreover, they are contradictory. Unfortunately, I still cannot recommend this manuscript for publication.

There are serious flaws in the work:

"In this study, RNA-seq Data were analyzed after comparison with reference genome. Due to incomplete genome annotation information, there are unannotated genes in intergenic regions. Moreover, transcripts containing intron information are identified due to alternative splicing. In addition, RNA-seq can detect the immature mRNA including intron information. "

Then why do you keep calling these unannotated genes "intergenic regions"? You should annotate these genes or at least make a suggestion.

Information about some mutations in an unknown part of the genome has no value.

"The upstream and downstream regions of genes separately means upstream sequence and downstream sequence of genes (default length: 5000 bp)."

Once again, why do you have these regions in your transcriptome?

"We found that 0.2 M NaHCO3 treatment is an appropriate concentration, which can trigger the stress response of birch seedlings, In addition, we found the more differentially expressed genes (DEGs) in birch plans under short time stress treatment were identified than that under long time stress treatment."

You should indicate where did you publish this data

"The open reading frame (ORF) of WRKY70 and NAC9 was cloned using In-Fusion ligase into the pROKII vector digested by Sma I, respectively. We added the contents in our manuscript (Line 405-407)."

You should also provide the reference to the vector map and sequence or include it in the manuscript, because the contents of the vector are unclear.

"In addition, in this part, the purpose of the expression analysis of DEGs to verify the RNA-seq data, so random selection of DEGs is reasonable. "

Once again, how two of these random genes (WRKY70 and NAC9) appeared to be those most responsive for saline-alkaline stress?

"The qRT-PCR analysis is relative expression of genes using the delta–delta CT method, which can not exactly calculate the gene expression"

As I already wrote, in this case it is measured in % compared to the reference gene expression. But you do not give any units of measurements at all (what do the numbers from -4 to 7 mean?)

" Now, we have uploaded the sequences of WRKY70 and NAC9 to NCBI and added their Genbank numbers to the part of 4.8 in our manuscript (Line 402-403)."

You should also upload RNA-seq results to a public database.

"DEGs encoding CYP82C2, cinnamyl-alcohol dehydrogenase (CAD) genes, and DREB,

WRKY, NAC, ERF, and bHLH TFs, were involved in response to stress, signal transduction, secondary metabolic process, regulation of jasmonic acid, and abiotic stimulus signaling pathway (Table S3)."

Were there any other gene families among DEGs?

"According to the hierarchical clustering analysis, the genes could be roughly classified into five groups."

What genes were classified to these groups? To which of these groups "DEGs encoding CYP82C2, cinnamyl-alcohol dehydrogenase (CAD) genes, and DREB, WRKY, NAC, ERF, and bHLH TFs" were classified?

Figure 4 is not informative. Why did you analyze only 100 DEGs? Where are the gene names on the map?

"Figure 4. Red and green in the heatmap indicate up-regulated and down-regulated genes, respectively."

What are numbers 3 and -3?? Units of measurements are required

"Figure 5."

The results described above do not comply with the figure. For example, you write that "the total

number of SNP loci in the intergenic regions ranged from 31,101 to34,383". However, on the figure there are 53965 mutations

Moreover, from this figure I can see that most of your mutations are not categorized at all. Why so? You have a reference genome, therefore there might be some information about the regions with mutations.

It looks that there is a difference in mutation number between control and treated samples which are supposed to be taken from the same plants. In which regions have you found these differences? How can you explain them?

"relative expression levels of these genes were consistent with the fragments per kilobase of exon per million fragments mapped ( FPKM) values"

"The expression of the WRKY70 and NAC9 TF genes was significantly up-regulated

under NaHCO 3 treatment compared with the control."

Were these two genes consistent with FPKM values or upregulated?

"The expression of the WRKY70 and NAC9 TF genes was significantly up-regulated

under NaHCO 3 treatment compared with the control."

"the relative expression of WRKY70 and NAC9 was approximately 300 times higher in WRKY70- and

NAC9-overexpressing plants than in pROKII-35S plants"

First of all, Figure 7 demonstrates that WRKY70 and NAC9 are not up-regulated under NaHCO 3 treatment in non-transformed plants (1% expression level remains unchanged).

If in control plants the expression of endogenous WRKY70 and NAC9 did not change in response to NaHCO3 stress, this means that these genes are not involved in stress response. Your results suggest that NaHCO3 affected only the transient expression (for example, the activity of A. tumefaciens). The expression of the gene under the control of a strong constitutive promoter also shouldn't have changed under stress treatment.

By the way, you probably should have introduced these genes to another plant species to be able to differentiate the expression of endogenous and exogenous gene.

"Figure 8. Analysis of ROS accumulation and cell membrane damage among transgenic and control

birch plants."

Why do you treat leaves if you studied the RNA from the roots?

"For example, three CADs involved in lignin biosynthesis were identified

(CAD1, CAD3, and CAD5). The expression of these three genes was mainly down-regu-

lated."

There is no information about it in Results section

"In our study, no differences

in the expression of genes encoding three DREB TFs in the 3-h NaHCO 3 treatment group

and control were observed; however, the expression of these genes was down-regulated

in the 6-h NaHCO 3 treatment relative to the control. Thus, the results of our study suggest

that DREB TFs are involved in complex regulatory pathways."

How exactly do you come to a conclusion that "DREB TFs are involved in complex regulatory pathways" based on the provided data? Which pathways?

Language became better